# De-Escalation Surgery in cT3-4 Breast Cancer Patients after Neoadjuvant Therapy: Predictors of Breast Conservation and Comparison of Long-Term Oncological Outcomes with Mastectomy

**DOI:** 10.3390/cancers16061169

**Published:** 2024-03-16

**Authors:** Corrado Tinterri, Erika Barbieri, Andrea Sagona, Alberto Bottini, Giuseppe Canavese, Damiano Gentile

**Affiliations:** 1Breast Unit, IRCCS Humanitas Research Hospital, Via Manzoni 56, 20089 Milan, Italy; corrado.tinterri@hunimed.eu (C.T.); erika.barbieri@cancercenter.humanitas.it (E.B.); andrea.sagona@cancercenter.humanitas.it (A.S.); alberto.bottini@cancercenter.humanitas.it (A.B.); giuseppe.canavese@cancercenter.humanitas.it (G.C.); 2Department of Biomedical Sciences, Humanitas University, Via Rita Levi Montalcini 4, 20090 Milan, Italy

**Keywords:** breast cancer, neoadjuvant therapy, breast-conserving surgery, mastectomy, de-escalation

## Abstract

**Simple Summary:**

Neoadjuvant therapy (NAT) has transformed the treatment of advanced breast cancer (BC), making previously inoperable tumors operable and allowing for the direct observation of treatment response. This approach reduces the need for extensive surgery, making breast-conserving surgery (BCS) a viable option for more patients. Despite the shift towards less-invasive surgery, the decision between BCS and mastectomy remains complex, influenced by numerous factors. Our study specifically looked at patients with cT3-4 BC treated with NAT, aiming to identify independent factors that predict the likelihood of undergoing BCS and to compare long-term oncological outcomes between BCS and mastectomy. We found that the absence of vascular invasion, smaller tumor size post-NAT, and achieving a complete response of the primary tumor were key predictors for breast conservation. Our results indicate that BCS post-NAT does not negatively impact long-term oncological outcomes, supporting its use as a safe option for patients with cT3-4 BC.

**Abstract:**

Background: Neoadjuvant therapy (NAT) has become increasingly employed for the treatment of cT3-4 breast cancer (BC), enabling breast-conserving surgery (BCS) in cases traditionally considered for mastectomy. This study aims to identify predictors for breast conservation post-NAT and to evaluate whether BCS influences long-term oncological outcomes. Methods: We retrospectively analyzed data from patients with cT3-4 BC who received NAT at the Breast Unit of IRCCS Humanitas Research Hospital, Milan, Italy, from October 2009 to April 2020. Surgical outcomes and long-term oncological results, such as disease-free survival (DFS), distant DFS (DDFS), overall survival (OS), and BC-specific survival (BCSS), were compared between the BCS and mastectomy groups. Results: Among 114 patients analyzed, 37 (32.5%) underwent BCS, and 77 (67.5%) had a mastectomy. The key predictors for opting for BCS included absence of vascular invasion, reduced tumor size post-NAT, and achieving ypT0 status. No significant differences in DFS, DDFS, OS, and BCSS were observed between the two surgical groups (log-ranks, *p* = 0.520, *p* = 0.789, *p* = 0.216, *p* = 0.559, respectively). Conclusions: BCS after NAT is a feasible and safe option for patients with cT3-4 BC, without adversely affecting long-term oncological outcomes. Identifying predictors of breast conservation can guide surgical decision-making, ensuring that patients receive optimal treatment.

## 1. Introduction

Neoadjuvant therapy (NAT)—i.e., systemic treatment administered prior to surgery—plays a crucial role in managing locally advanced breast cancer (BC). The application of pre-operative systemic therapy offers numerous potential advantages. First, it can transform inoperable cases into operable ones [1,2,3]. Secondly, NAT facilitates the in vivo evaluation of the tumor response to chemotherapy, which can inform treatment decisions post-surgery [4,5]. The response to NAT also holds prognostic significance, being closely linked to long-term oncological outcomes [4,6,7,8,9]. Furthermore, successful tumor reduction through NAT can minimize the extent of surgery required, eliminating the need for unnecessary axillary lymph node dissection (ALND) [10,11] and making breast-conserving surgery (BCS) possible for patients initially considered for mastectomy [12,13,14,15]. In the upfront surgical setting, BCS has been proven to be a safe alternative to mastectomy in terms of overall survival (OS) [16,17], with recent studies even suggesting superior survival rates associated with BCS [18,19]. Similar results were obtained in patients undergoing BCS following NAT, showing that breast conservation is not detrimental when compared to mastectomy [20,21,22], alongside offering enhanced cosmetic results and quality of life [23,24,25]. Thus, BCS after NAT is acknowledged as both feasible and safe. In a recent retrospective analysis, Gusic et al. [26] analyzed data on 149 patients with BC who received NAT to determine the rationale for mastectomy, especially in women eligible for BCS. A larger pathologic tumor size (2.05 cm versus 1.25 cm, *p* = 0.04) and histology (invasive lobular carcinoma versus invasive ductal carcinoma, *p* < 0.001) were associated with an increased rate of mastectomy. Despite this, BCS was successfully performed in 22 patients with cT3-4 BC. However, the shift from a planned mastectomy to BCS in patients responding to NAT remains less common, especially in cT3-4 tumors [23]. The surgical choice between BCS and mastectomy often remains a complex and multifaceted decision for breast surgeons, influenced by a variety of patient, tumor, and histopathological factors. Understanding these factors is vital for refining treatment approaches and aligning surgical decisions with the latest clinical evidence. We retrospectively collected and analyzed the data of patients with cT3-4 BC treated with NAT and subsequent surgery, either BCS or mastectomy. The aim of this study was to evaluate the differences between the two different surgical groups, identify the predictors of breast conservation in cT3-4 tumors, and compare the long-term oncological outcomes between BCS and mastectomy.

## 2. Materials and Methods

### 2.1. Study Design and Patient Management

We reviewed all the patients with cT3-4 BC who received NAT at the Breast Unit of IRCCS Humanitas Research Hospital in Milan, Italy, from October 2009 to April 2020. We collected information on patient age, menopausal status, tumor size and focality at baseline ultrasound (US) and/or magnetic resonance imaging (MRI), clinical stage of the tumor (cT3 or cT4), nodal involvement, subtype, histotype, presence of vascular invasion, and the specific NAT regimens used (Appendix A). The pre-operative loco-regional staging included bilateral mammography and US of both the breast and axilla. We did not exclude a priori patients with metastatic (cM1) disease. Rather, we specifically selected a group of oligometastatic patients with no comorbidities and a good performance status to receive pre-operative therapy. After experiencing a clinical–radiological response to systemic therapy, they became candidates for surgery, as part of a strategic, tailored approach. To evaluate the effectiveness of the systemic therapy and determine the size of the tumor prior to surgery, patients received an evaluation after completing NAT through various methods, which included bilateral US of the breast and axilla, MRI of the breast, or whole-body positron emission tomography (PET). A complete response of the primary tumor in the surgical specimen post-NAT was defined as ypT0. A pathologic complete response (pCR) was defined as no detectable invasive or non-invasive cancer in both the breast and nodal samples. The status of hormone receptors was ascertained through immunohistochemical (IHC) analysis, where a positive designation for both estrogen receptor (ER) and progesterone receptor (PgR) was given when reactivity was observed in over 1% of the cells [27]. The assessment of human epidermal growth factor receptor 2 (HER2) was performed using IHC and fluorescent in situ hybridization (FISH), with HER2 positivity defined by ASCO-CAP guidelines [28] as either an IHC score of 3+ or an IHC score of 2+ with FISH amplification. HER2-negative status was indicated by an IHC score of 1+ or 0. Subtypes were categorized based on hormone receptor and HER2 statuses. A multidisciplinary team comprising specialists in breast surgery, medical oncologists, pathologists, plastic surgeons, radiologists, and radiotherapists discussed the management for each cT3-4 BC patient upon diagnosis and following NAT. Treatment options included BCS or mastectomy. Exclusion criteria included patients who had undergone an excisional biopsy or debulking surgery as their initial BC treatment, male patients, those with a prior BC diagnosis or other cancers, cT1-2 tumors, unknown NAT regimen, disease progression during NAT, or a follow-up period of less than 50 months. Patients who failed to achieve a pCR and who had not undergone treatment with both anthracycline and taxane regimens prior to surgery were administered adjuvant chemotherapy based on taxane. For those with triple-negative and HER2-positive BC subtypes who did not reach pCR, capecitabine and trastuzumab emtansine (T-DM1) were given, respectively. Consent for surgery and the use of their clinical data was obtained from each patient.

### 2.2. Statistical Analysis

For the analysis of descriptive statistics, data on patients, including pre-surgery radiological assessment, tumor characteristics, surgical interventions, and follow-up information, were extracted from our institution’s database and are displayed as frequencies and percentages or medians and ranges. Subsequently, patients with cT3-4 BC were divided into two different surgical groups: BCS or mastectomy. To compare the means of continuous variables, we employed the independent samples *t*-test. For categorical variables, differences between the two surgical groups were assessed using the chi-square test (χ2 test) for variables with more than two categories or Fisher’s exact test for binary variables. In addition to univariate analyses, a multivariate logistic regression was performed to identify independent predictors for breast conservation in patients with cT3-4 BC treated with NAT. The multivariate analysis included any variable with significant correlation at univariate analysis (inclusion cut-off value *p* < 0.05). Disease-free survival (DFS) was defined as the duration between the BC surgery date and the first occurrence of any tumor, encompassing both loco-regional recurrence and distant metastasis. Distant DFS (DDFS) referred to the span from the BC surgery date until the identification of distant metastasis. OS was measured from the date of BC treatment to either death from any cause or the last known follow-up. BC-specific survival (BCSS) was defined as the time from BC treatment to death attributed directly to BC. The Kaplan–Meier method was employed to calculate the probabilities of recurrence and survival, while the log-rank test compared the two surgical cohorts of patients with cT3-4 BC who received NAT (BCS versus mastectomy). Updates to the follow-up data were made until 9 February 2024. A *p*-value of less than 0.05 was considered to indicate statistical significance. The analysis of data and creation of figures were conducted using IBM SPSS Statistics for Windows, Version 25.0. Armonk, NY, USA: IBM Corp.

## 3. Results

### 3.1. Patients’ Characteristics

Overall, 114 cT3-4 BC women who received NAT at the Breast Unit of IRCCS Humanitas Research Hospital (Milan, Italy) were included in our analysis. The median age was 50 years (range, 20–76), with 63 (55.3%) of the patients being post-menopausal. All patients underwent breast and axillary US as part of their pre-operative staging, while bilateral mammography, MRI of the breast, and PET were performed in 80 (70.2%), 40 (35.1%), and 57 (50.0%) patients, respectively. The median size of tumors before NAT was 51 mm (range, 11–115), with 89 (78.1%) patients presenting a single nodule. Regarding the stage before NAT, 73 (64.0%) were classified as cT3 and 41 (36.0%) as cT4, with 84 (73.7%) having nodal involvement (cN+). The majority received NAT with anthracycline and taxanes (*n* = 87, 86.1%), and 45 (39.5%) were treated with trastuzumab. The most common histologic subtype was luminal-like (*n* = 45, 39.5%), followed by HER2-positive (*n* = 49, 43.0%), and triple-negative (*n* = 20, 17.5%). After NAT, the median size of the tumors was 13.5 mm (range, 0–110). Overall, pCR was observed in 20.2% of the patients. The breast surgical treatment was BCS in 32.5% and mastectomy in 67.5% of the cases. Concerning axillary surgery, 37 (32.5%) of the patients received sentinel lymph node biopsy (SLNB) alone, without proceeding to ALND, while 77 (67.5%) patients were treated with ALND, either as initial treatment or following SLNB. In terms of post-operative treatment, 13 (11.4%) patients were given adjuvant chemotherapy, with 9 receiving a taxane-based regimen and 4 being treated with capecitabine. Adjuvant radiotherapy was administered to 95 (83.3%) of the patients. Endocrine therapy was provided to 72 (63.2%) patients, and T-DM1 was given to 33 (29.0%) patients. Descriptive statistics are summarized in Table 1.

### 3.2. Comparison of Characteristics between Surgical Groups (Breast-Conserving Surgery versus Mastectomy) and Predictive Factors of Breast Conservation

At the univariate analysis, age and menopausal status were evenly distributed (*p* = 0.570, and *p* = 0.857, respectively). The primary tumor’s stage was unbalanced in the two surgical groups, with a larger proportion of cT3 cancers among women treated with BCS (*n* = 29, 78.4%) compared to those receiving mastectomy (*n* = 44, 57.1%, *p* = 0.027). The mean pre- and post-NAT tumor size was larger in the mastectomy group than in the BCS group (57.7 mm versus 54.6 mm, *p* = 0.001, and 29.4 mm versus 13.0 mm, *p* = 0.004, respectively). ypT0 was more common in patients receiving BCS (*n* = 15, 40.5%) compared to those treated with mastectomy (*n* = 9, 11.7%, *p* < 0.001). Pre-operative radiological staging, subtype, and histotype distribution showed no statistically significant differences. At the multivariate analysis, three factors emerged as significantly related to the chance of receiving BCS. The absence of vascular invasion was associated with a higher probability of breast conservation [odds ratio (OR): 8.723, 95% confidence interval (95%CI): 0.232–0.440, *p* = 0.027]. A smaller tumor post-NAT was also a statistically significant predictor of BCS (OR: 6.811, 95%CI: 13.984–29.449, *p* = 0.011). Additionally, patients who achieved ypT0 after NAT more likely underwent BCS (OR: 15.470, 95%CI: 0.189–0.367, *p* < 0.001). The results of the univariate and multivariate analyses are summarized in Table 2.

### 3.3. Comparison of Long-Term Oncological Outcomes between Surgical Groups (Breast-Conserving Surgery versus Mastectomy)

After a median follow-up of 70.0 months (range, 52–185), 43 (37.7%) patients with cT3-4 BC who underwent NAT experienced a recurrence. Among these, 11 (9.7%) patients had only loco-regional recurrence, 19 (16.7%) patients had only distant recurrence, and 13 (11.4%) of the patients developed both loco-regional recurrences and distant metastases. Among those patients with cT3-4 BC receiving NAT, a total of 30 (26.3%) died; 24 of these deaths were directly attributed to BC, whereas 6 resulted from other causes. Specifically, in the different surgical groups, 6 out of 37 patients (16.2%) undergoing BCS and 24 out of 77 patients (31.2%) who had mastectomies died. Long-term oncological outcomes showed no significant difference in DFS, DDFS, OS, and BCSS between the BCS and mastectomy groups at the 3-, 5-, and 10-year rates. The 10-year DFS rates were 59.9% for BCS and 58.4% for mastectomy (*p* = 0.520). The 10-year DDFS rates were 60.9% for BCS and 59.5% for mastectomy (*p* = 0.789). The 10-year OS rates were 61.4% for BCS and 65.6% for mastectomy (*p* = 0.216). The 10-year BCSS rates were 61.4% for BCS and 73.3% for mastectomy (*p* = 0.559). These findings suggest that the type of surgical treatment, whether BCS or mastectomy, does not significantly impact the long-term oncological outcomes in patients with cT3-4 BC treated with NAT. A comparison of long-term oncological results is summarized in Table 3 and Figure 1 and Figure 2.

## 4. Discussion

The use of chemotherapy in the neoadjuvant setting has substantially increased in recent years [13,29]. Notably, this approach has greatly facilitated the shift from mastectomy to BCS as a surgical option, particularly in patients with initially diagnosed cT3-4 BC who were previously deemed unsuitable for breast conservation. Despite the substantial progress in systemic therapies that have improved the rates of complete response in primary tumors, recent neoadjuvant trials have reported lower rates of breast conservation [30]. This occurs even though most patients eligible for BCS after NAT often end up undergoing mastectomy [21,22,26,31,32,33,34]. In a large retrospective analysis involving 916 patients with BC treated with NAT, Li et al. [35] showed that patients with cT3 tumors were nearly six times more likely to undergo mastectomy than patients with cT1 tumors (OR: 5.74, 95%CI: 2.07–15.97, *p* = 0.003). Additionally, in a recent meta-analysis involving 36 studies and 12,311 patients with BC, Criscitiello et al. [36] demonstrated that achieving a complete response does not necessarily lead to an increased adoption of BCS in patients treated with NAT. In our study, we aimed to identify the predictors of breast conservation after NAT in cT3-4 tumors and to evaluate the long-term oncological outcomes, specifically in terms of recurrence and survival. This evaluation was conducted to determine whether opting for breast conservation after NAT compromises prognosis in any way.

According to multivariate analysis, three independent factors with a statistically significant correlation with the surgical choice after NAT were found to be important predictors of breast conservation in cT3-4 BC, including absence of vascular invasion, a smaller tumor, and achieving ypT0. The decision to proceed with mastectomy in patients with cT3-4 BC after NAT and potentially aggressive tumor features (vascular invasion, larger, and failure to achieve a complete response) is often driven by concerns over the risk of increased local recurrence and lower survival rates. Such concerns were highlighted by a 2018 meta-analysis conducted by the Early BC Trialists’ Collaborative Group (EBCTCG) [37], which reviewed ten randomized trials over a median period of nine years. This analysis revealed an increased rate of local recurrences at 10 and 15 years among patients who had BCS for tumors downsized by NAT in comparison to those who underwent BCS in the adjuvant setting for tumors of similar sizes (17.9% versus 13.2% and 21.4% versus 15.9%, respectively, log-rank *p* < 0.001). It is important to note that within these trials, two did not perform surgical removal post-NAT in instances of clinical complete response (Institut Bergonié Bordeaux and Institut Curie). Subgroup analyses showed that the risk of local recurrence after NAT compared to adjuvant treatment was greatest in these two trials (33.7% versus 20.4% after 10 years, log-rank *p* = 0.002) [38,39]. Moreover, there was a lack of information on axillary surgery and radiotherapy data.

On the other hand, our study suggests that the choice of surgical treatment, be it BCS or mastectomy, does not significantly influence the long-term oncological outcomes in patients with BC undergoing NAT, as corroborated by numerous other studies. Gwark et al. [20] retrospectively analyzed 1641 patients with BC who received NAT before surgery, of whom 839 (51.1%) underwent BCS + radiotherapy and 802 (48.9%) underwent mastectomy. Patients who underwent mastectomy had larger tumors (*p* < 0.001) and lower ypT0 rates (*p* = 0.005). For the breast conservation and mastectomy groups, the unadjusted 5-year DFS, DDFS, and OS rates were 87.0% and 73.1%, 89.5% and 77.0%, and 91.8% and 81.0%, respectively (all *p* < 0.001). Simons et al. [21] analyzed the results of 561 patients with BC treated with NAT, 362 (64.5%) with BCS and 199 (35.5%) with mastectomy. Mastectomy patients had larger tumors (*p* < 0.001). The unadjusted 5-year DFS was 90.9% for BCS versus 82.9% for mastectomy (*p* = 0.004). The unadjusted 5-year OS was 95.3% and 85.9% (*p* < 0.001), respectively. Moreover, in the BCS group, DFS and OS did not differ significantly between cT1, cT2, or cT3 tumors. Agrawal et al. [22] performed a retrospective study of 411 patients with non-metastatic, locally advanced BC who received NAT followed by surgery. The estimated 5-year DFS, DDFS, and OS rates of BCS and mastectomy groups were 63.9%, 71.0%, and 79.3% and 57.9%, 58.3%, and 71.5%, respectively. After adjusting for age, cT stage, cN stage, and radiotherapy, the BCS and mastectomy groups were found comparable in terms of DFS, DDFS, and OS. A recent meta-analysis [40] including 14 studies and 19,819 patients suggested that BCS after NAT is actually associated with significantly decreased risk of death (OR: 0.78, 95%CI: 0.69–0.89, *p* < 0.001), loco-regional recurrence (OR: 0.64, 95%CI: 0.48–0.85, *p* = 0.002), and DDFS (OR: 0.70, 95%CI: 0.53–0.94, *p* = 0.020) compared to mastectomy. Additionally, Werutsky et al. [41] conducted a pooled analysis of 10,075 primary patients with BC treated with NAT, revealing 5-year loco-regional recurrence rates of 7.8% in the BCS group and 11.3% in those undergoing mastectomy. In the I-SPY2 trial, a prospective, randomized study, Mukhtar et al. [42] evaluated the relationship between NAT response, assessed via the residual cancer burden (RCB) method, and loco-regional recurrence in 1462 patients with BC who received surgical treatment (BCS or mastectomy) from 2010 to 2021. With a median follow-up of 3.5 years, loco-regional recurrences were observed in 5.4% of BCS patients and 7.0% of mastectomy patients (*p* = 0.18). Patients with RCB 2/3 post-NAT had a notably reduced loco-regional free survival compared to those with RCB 0/1, irrespective of the surgical method. There was no significant difference in loco-regional-free survival between BCS and mastectomy patients with RCB 0/1 after NAT. Moreover, many other studies have consistently demonstrated that BCS does not compromise recurrence and survival rates in patients with BC treated with NAT [43,44,45,46,47].

It is certainly true that the reduction in tumor load by NAT presents a distinct challenge for breast surgeons in accurately locating and completely excising the primary tumor site. Despite this, breast surgeons must remember that patients with BC who are potential candidates for BCS should have the tumor(s) marked with a metallic clip before starting NAT [48,49]. The metallic clip remains detectable during the entire course of NAT, facilitating precise pre-operative localization of the tumor bed and intra-operative identification [50].

A significant limitation of our study is its retrospective design, which limits the ability to retrospectively evaluate patient preferences that significantly influence the surgical choice post-NAT. Factors such as fear of radiotherapy, the time commitment required for radiation treatment, and concerns over local recurrence significantly impact surgical decisions but were not quantitatively assessed in our analysis. Furthermore, considerations related to oncoplastic reconstruction, including delayed reconstruction options, were not explored in this study.

## 5. Conclusions

In conclusion, our study shows the feasibility and safety of BCS after NAT for cT3-4 patients with BC, challenging the traditional preference for mastectomy in this subgroup. We identified key predictors for opting for BCS over mastectomy, including the absence of vascular invasion, reduced tumor size post-NAT, and achieving ypT0. Importantly, our findings revealed that choosing BCS after NAT does not compromise long-term oncological outcomes compared to mastectomy. This supports the integration of BCS into treatment plans for appropriately selected patients, aligning surgical decisions with contemporary clinical evidence to enhance patient care.

## Figures and Tables

**Figure 1 cancers-16-01169-f001:**
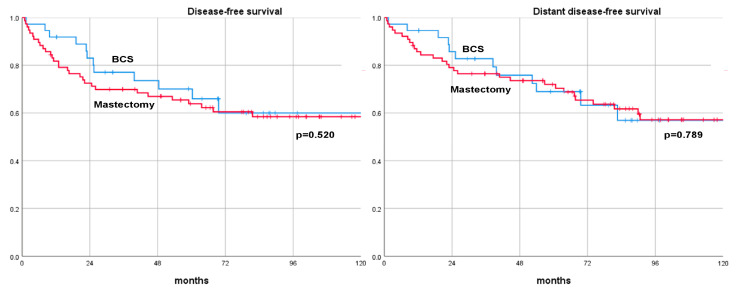
Disease-free survival and distant disease-free survival curves of patients with cT3-4 breast cancer treated with neoadjuvant therapy and surgery (breast-conserving surgery versus mastectomy). BCS: breast-conserving surgery.

**Figure 2 cancers-16-01169-f002:**
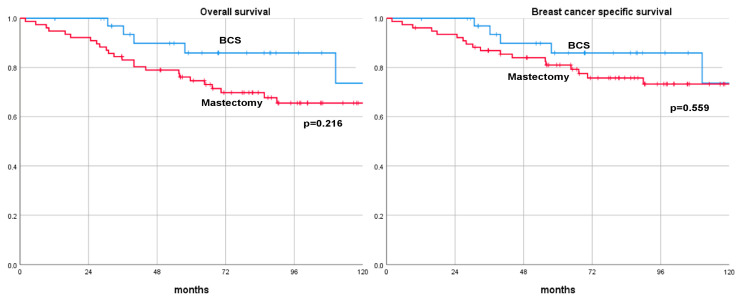
Overall survival and breast-cancer-specific survival curves of patients with cT3-4 breast cancer treated with neoadjuvant therapy and surgery (breast-conserving surgery versus mastectomy). BCS: breast-conserving surgery.

**Table 1 cancers-16-01169-t001:** Characteristics of 114 cT3-4 breast cancer patients treated with neoadjuvant therapy.

Characteristics	Number (%)/Median (Range)
Patients	
Age (years)	50 (20–76)
Post-menopausal	63 (55.3%)
Pre-operative staging	
Mammography	80 (70.2%)
Breast and axillary US	114 (100%)
MRI	40 (35.1%)
PET	57 (50.0%)
Size pre-NAT (mm)	51 (11–115)
Single nodule	89 (78.1%)
Stage pre-NAT	
cT3	73 (64.0%)
cT4	41 (36.0%)
cN0	30 (26.3%)
cN+	84 (73.7%)
cM1	19 (16.7%)
NAT with anthracycline only	21 (13.9%)
NAT with anthracycline and taxanes	87 (86.1%)
Trastuzumab	45 (39.5%)
Tumor	
Subtype	
Luminal-like	45 (39.5%)
HER2-positive	49 (43.0%)
Triple-negative	20 (17.5%)
Histotype	
Ductal	105 (92.1%)
Lobular	6 (5.3%)
Mucinous	3 (2.6%)
Vascular invasion	41 (36.0%)
pCR	23 (20.2%)
Size post-NAT (mm)	13.5 (0–110)
Stage post-NAT	
ypT0	24 (21.1%)
ypTis	7 (6.1%)
ypTmi	4 (3.5%)
ypT1a	8 (7.0%)
ypT1b	4 (3.5%)
ypT1c	25 (21.9%)
ypT2	20 (17.5%)
ypT3	11 (9.7%)
ypT4	11 (9.7%)
ypN0	61 (53.5%)
ypNmi	2 (1.8%)
ypN1	17 (14.9%)
ypN2	17 (14.9%)
ypN3	17 (14.9%)
Surgical treatment	
BCS	37 (32.5%)
Mastectomy	77 (67.5%)
SLNB not followed by ALND	37 (32.5%)
SLNB followed by ALND	10 (8.8%)
ALND	67 (58.7%)
Post-operative treatment	
Taxanes	9 (7.9%)
Capecitabine	4 (3.5%)
Radiotherapy	95 (83.3%)
Endocrine	72 (63.2%)
T-DM1	33 (29.0%)

Footnotes: US: ultrasound, MRI: magnetic resonance imaging, PET: positron emission tomography, NAT: neoadjuvant therapy, HER2: HER2 evaluated either on immunohistochemistry or on in situ hybridization, according to the ASCO CAP guidelines, pCR: pathologic complete response, BCS: breast-conserving surgery, SLNB: sentinel lymph node biopsy, ALND: axillary lymph node dissection, T-DM1: trastuzumab emtansine.

**Table 2 cancers-16-01169-t002:** Differences between surgical groups in cT3-4 breast cancer patients treated with neoadjuvant therapy.

Characteristics	BCS (No. 37) Tot. (%)/Mean (SD)	Mastectomy (No. 77) Tot. (%)/Mean (SD)	Univariate Analysis *p*-Value	Multivariate Analysis *p*-Value OR (95% CI)
Demographic				
Age (years)	51.1 (11.9)	52.4 (11.2)	0.570	-
Post-menopausal	20 (54.1)	43 (55.8)	0.857	-
Pre-operative staging				
Mammography	26 (70.3)	54 (70.1)	0.988	-
MRI	15 (40.5)	25 (32.5)	0.398	-
PET	14 (37.8)	43 (55.8)	0.072	-
Size pre-NAT (mm)	54.6 (13.6)	57.7 (25.0)	0.001 ^a^	0.508 0.442 (51.390–60.870)
Single nodule	24 (64.9)	65 (84.4)	0.059	-
Stage pre-NAT				
cT3	29 (78.4)	44 (57.1)	0.027 ^a^	0.138 2.249 (3.187–3.393)
cT4	8 (21.6)	33 (42.9)	-	-
cN+	29 (78.4)	55 (71.4)	0.430	-
cM1	4 (10.8)	15 (19.5)	0.245	-
Subtype				
Luminal-like	12 (32.4)	33 (42.9)	0.441	-
HER2-positive	19 (51.4)	30 (39.0)	-	-
Triple-negative	6 (16.2)	14 (18.1)	-	-
Histotype				
Ductal	35 (94.6)	70 (90.9)	0.698	-
Lobular	1 (2.7)	5 (6.5)	-	-
Mucinous	1 (2.7)	2 (2.6)	-	-
Histopathological				
Vascular invasion	8 (21.6)	33 (42.9)	0.027 ^a^	0.004 ^a^ 8.723 (0.232–0.440)
Size post-NAT (mm)	13.0 (17.2)	29.4 (38.9)	0.004 ^a^	0.011 ^a^ 6.811 (13.984–29.449)
ypT0	15 (40.5)	9 (11.7)	<0.001 ^a^	<0.001 ^a^ 15.470 (0.189–0.367)
Ki67 (%)	17.1 (17.2)	21.5 (23.3)	0.301	-

Footnote: BCS: breast-conserving surgery, SD: standard deviation, OR: odds ratio, 95% CI: 95% confidence interval, MRI: magnetic resonance imaging, PET: positron emission tomography, NAT: neoadjuvant therapy, HER2: HER2 evaluated either on immunohistochemistry or on in situ hybridization, according to the ASCO CAP guidelines, ^a^: statistically significant.

**Table 3 cancers-16-01169-t003:** Comparison of disease-free, distant disease-free, overall survival and breast-cancer-specific survival in cT3-4 breast cancer patients treated with neoadjuvant therapy.

Outcomes	BCS	Mastectomy	Log-Rank Test
DFS rate			0.520
3-year	77.1%	69.9%	
5-year	65.9%	63.9%	
10-year	59.9%	58.4%	
DDFS rate			0.789
3-year	82.8%	76.4%	
5-year	69.0%	70.4%	
10-year	60.9%	59.5%	
OS rate			0.216
3-year	93.4%	83.1%	
5-year	85.9%	74.7%	
10-year	61.4%	65.6%	
BCSS rate			0.559
3-year	93.4%	86.9%	
5-year	85.9%	81.0%	
10-year	61.4%	73.3%	

Footnotes: BCS: breast-conserving surgery, DFS: disease-free survival, DDFS: distant disease-free survival, OS: overall survival, BCSS: breast-cancer-specific survival.

## Data Availability

Data supporting reported results can be found in the Appendix A.

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
