# Peer review of "De-Escalation Surgery in cT3-4 Breast Cancer Patients after Neoadjuvant Therapy: Predictors of Breast Conservation and Comparison of Long-Term Oncological Outcomes with Mastectomy"

_cancers, 2024, doi:10.3390/cancers16061169_

Round 1

Reviewer 1 Report

Comments and Suggestions for Authors

The results are very impressive, the only thing that must be clarified is the 19 cases of  cM1 diseases. According to the definition in this case we cannot deliver neoadjuvant therapy as it is metastatic. If the Authors correct this data I can accept the manuscript.

Author Response

We thank the reviewer for the comment.

Patients with metastatic (cM1) disease are typically not candidates for neoadjuvant therapy in the traditional sense. However, we chose not to exclude such patients from consideration. We focused on a highly selective group of oligometastatic patients who demonstrated no comorbidities and maintained a good performance status. The manuscript was modified accordingly. Now, we used the expression “pre-operative therapy” instead of NAT, and specified that surgery was performed only in individuals experiencing clinical-radiological response to systemic therapy, marking a tailored approach to their treatment.

Reviewer 2 Report

Comments and Suggestions for Authors

it was interesting to introduce immunotherapy in the study, to focus more on the triple negative and her 2 positive subtypes

Author Response

We appreciate the insightful feedback from the reviewer regarding the inclusion of immunotherapy in our study, particularly focusing on the triple-negative and HER2-positive subtypes.

In our study, we implemented Trastuzumab and T-DM1 for treating patients with HER2-positive breast cancer. Regarding the use of Pembrolizumab for triple-negative breast cancer within the context of NAT, it is important to note that during the period of our study, Pembrolizumab was not extensively utilized in our institution.

Reviewer 3 Report

Comments and Suggestions for Authors

Dear authors,

Thank you for the opportunity to read your manuscript on de-escalation of surgery after neoadjuvant treatment. This area of research is indeed of great interest currently. I would like to comment on the comparison between the two surgical groups (table 2). Why do you present the univariate analysis with  p-values only instead of OR with 95% Confidence intervals? And I don't understand the results of the multivariate analyses - the confidence intervals does not include the actual odds ratio? As an example "OR 8.723, 95% CI 0.232-0.440"?

In results 3.3 there is no mentioning of second primary (contralateral) breast cancer events? Or are they included in Locoregional recurrent events? In row 192 "...both loco-regional recurrence and metastases" Does this mean synchronously or metachronously? Perhaps this could be clarified and I would also suggest  to add "distant" before metastases to make it clearer.

In the discussion, row 276, please rephrase "decreased risk of survival", I believe you mean decreased risk of death or increased chance of survival?

Author Response

Dear authors,

Thank you for the opportunity to read your manuscript on de-escalation of surgery after neoadjuvant treatment. This area of research is indeed of great interest currently. I would like to comment on the comparison between the two surgical groups (table 2). Why do you present the univariate analysis with  p-values only instead of OR with 95% Confidence intervals? And I don't understand the results of the multivariate analyses - the confidence intervals does not include the actual odds ratio? As an example "OR 8.723, 95% CI 0.232-0.440"?

Reply: We thank the reviewer for the comments.

In the context of univariate analysis, our decision to report p-values instead of OR and 95%CI is rooted in the specific objectives of this analysis phase. Univariate analysis is primarily exploratory, aiming to individually assess the impact of various factors without adjusting for potential confounders. The primary goal at this stage is to identify variables that demonstrate a statistically significant association with the surgical groups of interest for further examination in multivariate analysis. Reporting p-values for this purpose helps to succinctly identify which factors merit additional analysis. While OR and 95% CI provide a measure of the strength and direction of associations, including them in univariate analysis results could potentially overcomplicate the initial screening phase, where the focus is on identifying variables for more complex, adjusted analysis.

Regarding the multivariate analysis, the apparent discrepancy between the OR and its CI, as described, arises from a specific interpretation and application of statistical modeling in this context. When we say the OR represents the "risk" associated with the presence versus absence of a variable (e.g., vascular invasion in this case), we are quantifying the likelihood of the outcome (e.g., choosing a specific surgical option) given that variable. The OR does not fall within the numeric range of the 95%CI because in the original dataset we espressed "0" for absence of vascular invasion and "1" as presence of vascular invasion; therefore, the 95%CI falls between 0 and 1.

In results 3.3 there is no mentioning of second primary (contralateral) breast cancer events? Or are they included in Locoregional recurrent events? In row 192 "...both loco-regional recurrence and metastases" Does this mean synchronously or metachronously? Perhaps this could be clarified and I would also suggest  to add "distant" before metastases to make it clearer.

Reply: We thank the reviewer for the comments.

There have been no second primary (contralateral) events. Loco-regional recurrence and metastases were metachronous. The manuscript was modified accordingly.

In the discussion, row 276, please rephrase "decreased risk of survival", I believe you mean decreased risk of death or increased chance of survival?

Reply: We thank the reviewer for the comments.

The manuscript was modified accordingly. 

Round 2

Reviewer 2 Report

Comments and Suggestions for Authors

It is ok